# Three-Dimensional Numerical Study of Hydrodynamic Interactions between Pectoral Fins and the Body of Aquatic Organisms

**DOI:** 10.3390/biomimetics9030156

**Published:** 2024-03-01

**Authors:** Kotaro Morifusa, Tomohiro Fukui

**Affiliations:** 1Department of Master’s Program, Kyoto Institute of Technology Matsugasaki Goshokaido-cho, Sakyo-ku, Kyoto 606-8585, Japan; m2623033@edu.kit.ac.jp; 2Department of Mechanical Engineering, Kyoto Institute of Technology Matsugasaki Goshokaido-cho, Sakyo-ku, Kyoto 606-8585, Japan

**Keywords:** biomimetics, pectoral fins, flapping amplitude, flapping frequency, cyclic swimming

## Abstract

Fish swimming has attracted attention as a locomotion system with excellent propulsive efficiency. They swim by moving their body, fins, and other organs simultaneously, which developed during evolution. Among their many organs, the pectoral fin plays a crucial role in swimming, such as forward–backward movement and change of direction. In order to investigate the hydrodynamic interaction between pectoral fins and fish bodies, we examined the asymmetric flapping motion of the pectoral fin concerning the body axis and investigated the effect of the pectoral fin on the propulsive performance of the body of a small swimming object by numerical simulation. In this study, the amplitude ratio, frequency ratio, and phase of the body and pectoral fin varied. Therefore, although propulsive performance increased in tandem with the frequency ratio, the amplitude ratio change had negatively affected the propulsive performance. The results revealed that the propulsive performance of the fish was high even in low-frequency ratios when the phase difference was varied. The highest propulsion efficiency increased by a factor of about 3.7 compared to the phase difference condition of 0.

## 1. Introduction

In recent years, robots have been developed in various fields. These include underwater robots which require development and improvement of their efficiency for surveying unexplored areas of oceans. There are three main types of underwater robots: manned submersibles (HOV: Human Occupied Vehicles) that are directly operated by a person on board, remotely operated vehicles maneuvered by a person from outside of a cable, and autonomous underwater vehicles (AUVs), which are fully automated. Among them, AUVs can survey a wide area because they do not require cables for power transmission, and they can survey harsh environments, such as the Arctic Ocean [1]. Although this propulsion system has high energy efficiency, screw propellers have been damaging the surrounding environment due to their sharpness. Therefore, as a new propulsion mechanism to replace the screw propeller, the swimming mechanism of aquatic organisms that can adapt to the underwater environment has been attracting attention.

The swimming mechanism has been adopted as a propulsion mechanism in recent years because of the excellent characteristics of aquatic organisms, such as high swimming efficiency, rapid acceleration, deceleration [2], and turning performance [3]. In contrast, the swimming mechanism of aquatic organisms is complex and delicate, and it is difficult to perform detailed experiments using living organisms due to individual differences. Swimming methods of aquatic organisms vary greatly depending on their growth process and habitat.

Aquatic organisms have evolved in various ways to achieve better propulsion mechanisms, and the fins are among the organs that have developed during evolution and have been a key feature of fish evolutionary diversification [4,5]. Among them, the pectoral fin plays numerous roles such as braking, turning, and balance maintenance during locomotion [6,7,8]. Webb et al. [9] reported that the pectoral fin generates a cutting force for postural control and discussed its role in improving postural stability. In addition, while many pectoral fin studies have focused on rays [10,11], recent years have shown the effects of pectoral fins in a wide variety of aquatic organisms, including dolphins and turtles [12,13]. The pectoral fins play an auxiliary role in propulsion by flexion of the body to achieve maneuverability such as a change of direction and postural stability, but they also serve as an organ that further enhances the propulsive performance of aquatic organisms. The pectoral fins contribute to propulsion velocity and pectoral fin vortices reduce drag on the body [14,15]. Recently, the effect of pectoral fin flexibility on swimming was reported [16,17,18,19], and the pectoral fin is expected to improve the propulsive performance of aquatic organisms.

Therefore, studies on fish swimming have been conducted through simulations and experiments. However, in many previous studies, the left and right pectoral fins always move ideally symmetrically concerning the body axis, and few simulations have been conducted under conditions where the pectoral fins move asymmetrically on both sides. Li et al. [20] found in their three-dimensional analysis that the pectoral fin changed the behavior of rotational motion when the pectoral fins performed an asymmetric motion on the left and right. The effects of the phase difference between the left and right pectoral fins on swimming in rays and manta rays have also been reported [21,22]. From this point, the hydrodynamic interaction between the pectoral fins and the body is considered to be working in the out-of-synchronized motion of the body and the pectoral fins, resulting in complex swimming of the fish. In this respect, understanding the asymmetry of the pectoral fins and the correlation between the phase of the body and the pectoral fins and the propulsive performance of the fish will provide new insights into improving the performance of underwater robots that mimic aquatic organisms.

Fish are optimized not only in terms of shape, but also in terms of locomotion so that they emit the most effective signals in response to the fluid environment. Therefore, the objective of this study was to investigate the relationship between pectoral fin motion and the overall propulsive performance of fish considering asymmetry and to identify effective swimming methods in which the interaction between the pectoral fins and the body contributes the most to improving propulsive performance. This study aims to improve the performance of a robot that mimics fish swimming by investigating the relationship between fish locomotion and swimming performance. In this study, a simple model of the pectoral fin and body was applied to evaluate the propulsive performance of the fish when the amplitude and frequency ratios of the pectoral fin and the phase between the pectoral fin and body were varied. This study mentioned the effect of the phase difference between the body and the pectoral fin on the propulsive performance, which is unique in that it evaluates the swimming performance by the change of the phase difference.

## 2. Materials and Methods

### 2.1. Materials

In this paper, the swimming of fish was reproduced not by an experiment but by three-dimensional computational fluid dynamics. We simulated the fluid flow and reproduced the motion of the fish by using the flow chart shown in Figure 1.

### 2.2. Fish Model and Dynamics

Figure 2 illustrates the model outline. Here, the length of the rigid plate reproducing the body was set as the characteristic length *L*, and the length of the pectoral fin, *L*_fin_, was set at *L*/4. In this study, the width of the pectoral fin, *L*_s,fin_, was set to the same length as *L*_fin_. The pectoral fins were placed 0.2 *L* from the head of the body.

The body’s oscillating motion was represented by the following equation for the computational model: In this study, a motion was performed in the *x*–*y* plane, and no motion in the *z* direction was assumed. As in the previous study [23], the body’s oscillating motion was simplified as the pitching motion shown in Equation (1). In this study, the computational model performed only rotational motion with the head coordinates fixed.
(1)$$\theta_{\mathrm{body}}\left(t\right)=\theta_{\mathrm{body},\mathrm{amp}}\cos\left(2\pi f_{\mathrm{body}}t\right)$$
where $f_{\mathrm{body}}$ is the frequency of the body. Here, the oscillation angle $\theta_{\mathrm{body},\mathrm{amp}}$ was given by the following equation:(2)$$\theta_{\mathrm{body},\mathrm{amp}}=\sin^{-1}\left(A_{\mathrm{body}}\;/\;L\right),$$
where $A_{\mathrm{body}}$ is the amplitude of the body and *L* is the length. $A_{\mathrm{body}}\;/L$ is the amplitude ratio of the body, and in this study, the amplitude ratio was fixed at 0.1. The swimming motions of the right pectoral fin shown in Figure 3a was expressed using the following equation:(3)$$\theta_{\mathrm{fin}\_r}\;\left(t\right)=-\theta_{\mathrm{fin},\mathrm{amp}}\cos\left(2\pi f_{\mathrm{fin}}t\;+\varphi \right)+\theta_{\mathrm{fin}0}.$$

The swimming motions of the left pectoral fin were expressed using the following equation:(4)$$\theta_{\mathrm{fin}\_l}\;\left(t\right)=\theta_{\mathrm{fin},\mathrm{amp}}\cos\left(2\pi f_{\mathrm{fin}}t\;+\varphi \;+\varphi_{\mathrm{fin}}\right)-\theta_{\mathrm{fin}0},$$
where *φ* is the phase difference between the body and the pectoral fin, and $\varphi_{\mathrm{fin}}$ is the phase difference between the left and right pectoral fins. $\varphi_{\mathrm{fin}}$ was set to 180° when the pectoral fin and the body of the fish moved at the same frequency and 0° when the pectoral fin frequency was higher than the fish body frequency. $\theta_{\mathrm{fin}0}$ is the angle between the center of vibration of the pectoral fin and the body, respectively. The red line in Figure 3b represents the body plate. As shown in Figure 3b, the angle between the pectoral fin and the body was set to reach 20° during the pectoral fin’s internal rotation. The oscillation angle $\theta_{\mathrm{fin},\mathrm{amp}}$ is shown below. $A_{\mathrm{fin}}$ corresponds to the amplitude of the pectoral fin, $L_{\mathrm{fin}}$ is the length of the pectoral fin, and $A_{\mathrm{fin}}\;/\;L_{\mathrm{fin}}$ is the amplitude ratio of the pectoral fin.
(5)$$\theta_{\mathrm{fin},\mathrm{amp}}=\sin^{-1}\left(A_{\mathrm{fin}}\;/\;L_{\mathrm{fin}}\right).$$

The Reynolds number was defined by this equation in the swimming motion.
(6)Re=ULν,
where *U* is the average velocity of the body of the tail of the fish and *ν* is the kinematic viscosity of the surrounding fluid. *U* can be obtained as follows:(7)$$U\;=2L\theta_{\mathrm{body},\mathrm{amp}}f_{\mathrm{body}}.$$

In this study article, we simulate under the conditions of *L* = 10.0 mm, *U* = 50.0 mm/s, *ν* = 1.0 × 10^−6^ m^2^/s, and *Re* = 500.

### 2.3. Regularized Lattice Boltzmann Method

The regularized lattice Boltzmann method [24,25] was used as the governing equation of the fluid. The method is designed to reduce the calculation cost by modifying an algorithm in the original lattice Boltzmann method. The lattice Boltzmann equation takes the form below [26].
(8)$$f_{\alpha }\left(t+\delta t,\;x+e_{\alpha }\delta t\right)=f_{\alpha }\left(t,\;x\right)+\frac{1}{\tau }\left(f_{\alpha }^{\mathrm{eq}}\left(t,\;x\right)–\;f_{\alpha }\left(t,\;x\right)\right).$$

Furthermore, the distribution function *f_α_* can be expanded in terms of moments of discrete velocity moments up to the second-order moment component as follows:(9)$$f_{\alpha }\;≅\;w_{\alpha }\left(a+b_{i}e_{\alpha i}+c_{\mathrm{ij}}e_{\alpha i}e_{\alpha j}\right)$$
where *a*, *b_i_*, and *c_ij_* are constants, $w_{\alpha }$ is the weight function, and $e_{\alpha i}$ $e_{\alpha j}$ $e_{\alpha k}$ are each component values of the discrete velocity vector $e_{\alpha }$. In this study, the 3D27V lattice speed model was applied, and the weight function $w_{\alpha }$ and the discrete speed vector $e_{\alpha }$ are expressed as follows:(10)$$w_{\alpha }=\left\{\begin{array}{c} 8/27\\ 2/27\\ \begin{array}{c} 1/54\\ 1/216 \end{array} \end{array}\begin{array}{c} \left(\alpha =0\right)\\ \left(\alpha =1\sim 6\right)\\ \begin{array}{c} \left(\alpha =7\sim 18\right)\\ \left(\alpha =19\sim 26\right), \end{array} \end{array}\right.$$
(11)$$e_{\alpha }=\left\{\begin{array}{c} c\left(0,\;0,\;0\right)\\ c\left(\pm 1,\;0,\;0\right),c\left(0,\;\pm 1,\;0\right),c\left(0,\;0,\;\pm 1\right)\\ \begin{array}{c} c\left(\pm 1,\;\pm 1,\;0\right),\;c\left(\pm 1,\;0,\;\pm 1\right),\;c\left(0,\;\pm 1,\;\pm 1\right)\\ c\left(\pm 1,\;\pm 1,\;\pm 1\right) \end{array} \end{array}\right.\begin{array}{c} \left(\alpha =0\right)\\ \left(\alpha =1\sim 6\right)\\ \begin{array}{c} \left(\alpha =7\sim 18\right)\\ \left(\alpha =19\sim 26\right), \end{array} \end{array}$$
where *c* is the advection velocity. The macroscopic physical quantities, density *ρ*, and momentum *ρ**u*** are as follows [27]:(12)$$\rho =\sum_{\alpha }f_{\alpha },$$
(13)$$\begin{array}{c} \rho u=\sum_{\alpha }f_{\alpha }e_{\alpha }. \end{array}$$

Stress tensor at the nonequilibrium part can be obtained as follows [24]:
(14)$${\Pi_{\mathrm{ij}}}^{\mathrm{neq}}=\sum_{\alpha }{f_{\alpha }e}_{\alpha i}e_{\alpha j}-\frac{c^{2}}{3}\rho \delta_{\mathrm{ij}}-\rho u_{i}u_{j}.$$

Equation (9) can be expressed using the equilibrium distribution function $f_{\alpha }^{\mathrm{eq}}$ and the nonequilibrium part of the distribution function $f_{\alpha }^{\mathrm{neq}}$ as follows:(15)$$f_{\alpha }=f_{\alpha }^{\mathrm{eq}}+f_{\alpha }^{\mathrm{neq}}.$$

The equilibrium distribution function $f_{\alpha }^{\mathrm{eq}}$ [27] and the nonequilibrium part of the distribution function $f_{\alpha }^{\mathrm{neq}}$ [24] are expressed as follows:(16)$$f_{\alpha }^{\mathrm{eq}}=w_{\alpha }\rho \left\{1+\frac{3\left(e_{\alpha }\cdot u\right)}{c^{2}}+\frac{9{\left(e_{\alpha }\cdot u\right)}^{2}}{2c^{4}}\;–\frac{3u^{2}}{2c^{2}}\right\},$$
(17)$$f_{\alpha }^{\mathrm{neq}}=\frac{9w_{\alpha }}{2c^{2}}\left(\frac{e_{\alpha i}e_{\alpha j}}{c^{2}}-\frac{1}{3}\delta_{\mathrm{ij}}\right)\Pi_{\alpha }^{\mathrm{neq}}$$

Therefore, time development $f_{\alpha }$ can be expressed by the following equation.
(18)$$f_{\alpha }\left(t+\delta t,\;x+e_{\alpha }\delta t\right)=f_{\alpha }^{\mathrm{eq}}\left(t,x\right)+\left(1\;–\;\frac{1}{\tau }\right)f_{\alpha }^{\mathrm{neq}}\left(t,x\right).$$

Thus, the lattice Boltzmann BGK (Bhatnagar–Gross–Krook) equation with the incompressible formulation and the equilibrium distribution function of pressure, $p_{\alpha }^{\mathrm{eq}}$, are obtained by the following equations:(19)$$p_{\alpha }\left(t\;+\;\delta t,\;x\;+e_{\alpha }\delta t\right)=p_{\alpha }^{\mathrm{eq}}\left(t,\;x\right)+\left(1-\frac{1}{\tau }\right)p_{\alpha }^{\mathrm{neq}}\left(t,\;x\right),$$
(20)$$p_{\alpha }^{\mathrm{eq}}=w_{\alpha }\left[p+\rho_{0}\left\{\left(e_{\alpha }\cdot u\right)+\frac{3{\left(e_{\alpha }\cdot u\right)}^{2}}{2c^{2}}–\frac{u^{2}}{2}\right\}\right].$$

Using the pressure distribution function $p_{\alpha }$ calculated above, the macroscopic physical quantities, the pressure *p*, and the flow velocity vector ***u***, are as follows:(21)$$\begin{array}{c} p=\sum_{\alpha }p_{\alpha }, \end{array}$$
(22)$$\begin{array}{c} u=\frac{1}{\rho_{0}{c_{s}}^{2}}\sum_{\alpha }p_{\alpha }e_{\alpha }. \end{array}$$

### 2.4. Virtual Flux Method

In this study, the virtual flux method [28] was used to represent objects of arbitrary shape on a Cartesian grid. The virtual flux method can calculate flow phenomena around and inside an arbitrarily shaped object without adding external forces. This method is easy to implement because it requires only the addition of a virtual flux calculation routine to the usual flow field calculation program, and only the numerical flux is changed without modifying the computational grid. It also has the advantage of capturing the pressure field around the object more sharply than other calculation methods, such as the immersed boundary method. Furthermore, it has the advantage of superior computational efficiency compared to the immersed boundary method [29].

A schematic view of the virtual boundary points in the virtual flux method is shown in Figure 4. Virtual boundary points were placed at the intersection of the object surface and the discrete velocity direction, as shown by the white dots in Figure 4.

The boundary conditions for the pressure and velocity vector on the virtual boundary points are displayed below.
(23)$$\frac{\partial p_{\mathrm{vb}}}{\partial n}=0,$$
(24)$$u_{\mathrm{vb}}=u_{\mathrm{wall}}.$$
where $p_{\mathrm{vb}}$ is the pressure on the virtual boundary point, $u_{\mathrm{vb}}$ is the velocity vector on the virtual boundary point, and $u_{\mathrm{wall}}$ is the velocity vector on the arbitrarily shaped object wall.

First, the pressure $p_{\mathrm{vb}}$ at the virtual boundary point vb, as shown in Figure 5, is obtained. Pressure is evaluated by approximating the boundary condition of pressure with second-order accuracy using Equation (25). $p_{1}$ and $p_{2}$ are the pressures at the points *h*_1_ and *h*_2_ away from the virtual boundary point vb in the normal direction of the virtual boundary plane, as shown in Figure 5. In this study, *h*_1_ = $\sqrt{3}$ *δx* and *h*_2_ = 2$\sqrt{3}$ *δx* were applied to the three-dimensional analysis when the distance between the lattices was *δx*.
(25)$$p_{\mathrm{vb}}=\frac{{h_{2}}^{2}p_{1}\;–{h_{1}}^{2}p_{2}}{{h_{2}}^{2}–{h_{1}}^{2}}.$$

The flow field separated into fluid and virtual objects by the virtual boundary surface is shown in Figure 6. The virtual physical quantity *q*_D_^*^ at grid point D is obtained by linear extrapolation from the physical quantities at grid point C and the virtual boundary point vb. *Q*_D_^*^ is obtained by Equation (26) when *a* is 0.5 or greater. However, if the value of *a* is extremely small, the denominator of Equation (28) becomes negligibly small, and the calculation becomes unstable. In order to avoid this, when *a* is less than 0.5, the physical quantity is interpolated using grid point E instead of grid point C, as shown in Equation (27).
(26)$${Q_{D}}^{*}=-\frac{b}{a}q_{C}+\frac{1}{a}q_{\mathrm{vb}}\left(a\geq 0.5\right),$$
(27)$${q_{D}}^{*}=-\frac{b^{*}}{a^{*}}q_{E}+\frac{1}{a^{*}}q_{\mathrm{vb}}\left(a<0.5\right).$$

From the virtual physical quantity *q*_D_^*^ obtained from these equations, a virtual equilibrium distribution function can be calculated.
(28)$${P_{\alpha }}^{\mathrm{eq}*}(t,\;x_{D})=w_{a}\left[{p_{D}}^{*}+\rho_{0}\left\{\left(e_{a}\cdot {u_{D}}^{*}\right)+\frac{3{\left(e_{a}\cdot {u_{D}}^{*}\right)}^{2}}{2c^{2}}-\frac{{{u_{D}}^{*}}^{2}}{2}\right\}\right].$$

The nonequilibrium component of the virtual distribution function $p_{\alpha }^{\mathrm{neq}}$ is calculated from the information near the virtual boundary points as follows:(29)$$p_{\alpha }^{\mathrm{neq}}\left(t,\;x_{\mathrm{vb}}\right)=p_{\alpha }^{\mathrm{neq}}\left(t,\;x_{C}\right)\left(a\;\geq \;0.5\right),$$
(30)$$p_{\alpha }^{\mathrm{neq}}\left(t,\;x_{\mathrm{vb}}\right)=p_{\alpha }^{\mathrm{neq}}\left(t,\;x_{E}\right)\left(a\;<\;0.5\right).$$

Using the virtual distribution function at grid point D and the nonequilibrium part of the distribution function calculated above, the distribution function at the next time step at grid point C can be calculated as follows.
(31)$$P_{\alpha }\left(t+\delta t,\;x+e_{\alpha }\delta t\right)=p_{\alpha }^{\mathrm{eq}}\left(t,x_{D}\right)+\left(1-\frac{1}{\tau }\right){p_{\alpha }^{\mathrm{neq}}}^{*}\left(t,x_{D}\right).$$

## 3. Validation and Verification

First, a single 3D flapping plate analysis was simulated to verify the reliability of the 3D analysis code, followed by verification of the appropriate grid resolution of the computational model. The effects of the amplitude ratio, frequency ratio, and phase difference in propulsion performance were considered.

### 3.1. Validation Problem of the Three-Dimensional Vibrating Plate

Here, for a plate placed parallel to the *z*-axis, the oscillatory motion is expressed by the following equation: The thickness of the plate was not considered.
(32)$$\begin{array}{c} X\left(t\right)=L\sin\left(2\pi \mathrm{ft}\right), \end{array}$$
(33)$$\begin{array}{c} \theta \left(t\right)=\frac{\pi }{2}-\frac{\pi }{4}\sin\left(2\pi \mathrm{ft}+\frac{\pi }{3}\right), \end{array}$$
where *x*(*t*) is the displacement of the *x*-coordinate of the center of gravity, *θ*(*t*) is the vibration angle, and *f* is the frequency. Equation (32) represents vibration in the translational direction (heaving motion) and Equation (33) represents vibration against rotation (flapping motion). The length of the plate *L* was defined as the characteristic length, and the average vibration velocity *U* in the translational direction was defined as the characteristic velocity. In this study, *L* = 10.0 mm, *U* = 50.0 mm/s were applied as mentioned in Section 2.1. The frequency *f* was calculated from the average velocity *U* in the translational direction. The length of the plate in the *z*-axis direction was 4*L*. The history of motion is shown in Figure 7.

Figure 8 shows the computational domain and the arrangement of the blocks. The four-tier multiblock method [30,31] was applied, and for the multiblock method, the blocks were designated as Block 1, Block 2, Block 3, and Block 4, starting from the block with the highest lattice resolution. The lattice widths *δx*_1_, *δx*_2_, *δx*_3_, *δx*_4_ for each block satisfied *δx*_2_ = 2*δx*_1_, *δx*_3_ = 2*δx*_2_, *δx*_4_ = 2*δx*_3_. The simulation area was set to 20 *L* × 20 *L* × 16 *L*. The plate was placed so that its center of gravity was at the initial position (10 *L*, 10 *L*, 8 *L*) in the region, as shown in Figure 8a. In the *x*–*y* plane, the pressure was fixed at a constant value of 1/3, and the flow velocity was assumed to have a normal direction gradient of 0. The boundary conditions between the plate and the surrounding fluid are described in Section 2.3. For the initial conditions, the pressure was set at 1/3, and the flow velocity at 0. In this study, the Reynolds number was set to *Re* = 100. In order to confirm the numerical reliability of the results, the number of grid cells per characteristic length was set to 16, 32, 64, and 128 cells. The lift coefficient *C*_L_, given by the following equation, was used to evaluate the physical quantities in the flow around the plate.
(34)$$C_{L}=\frac{F_{y}}{\frac{1}{2}\rho U^{2}S_{p}}.$$
where *S_p_* is the surface area of the plate and *F_y_* is the fluid force acting in the *y* direction. Due to the large influence of the impact departure immediately after the start of the calculation, seven cycles of motion were performed, and the value of the lift coefficient *C*_L_ at the last cycle was used for the evaluation.

Figure 9 shows a comparison of the lift coefficient *C*_L_ as the number of grids per representative length varied. This figure shows the time history of the lift coefficient *C*_L_ was similar for each grid resolution. As the number of grids per representative length increased, the high-frequency oscillations decreased and converged to a certain waveform. These high-frequency oscillations were nonphysical oscillations caused by objects crossing the lattice, and the effect could be reduced by increasing the lattice resolution. These results confirm the numerical reliability of the code. Figure 10 shows a comparison of the lift coefficient *C*_L_ among calculations when the number of cells per characteristic length is 128. This figure discovered that the time history of the lift coefficient *C*_L_ was similar to that in previous studies [32,33].

Figure 11 shows the vorticity distribution around the *z*-axis in the *x–y* plane, which is a plane that passes through the center of gravity of the plate in the last cycle. A strong vortex was generated along the plate at the beginning of the stroke at *t* = 6.8*T* (Figure 11a) and that vortex became larger at *t* = 6.0*T* (Figure 11b) and was peeled off between strokes at *t* = 6.2*T* (Figure 11c), confirming the validity of the results obtained from this analysis. Additionally, these results confirmed the reliability of the calculation code using the regularized lattice Boltzmann method, virtual flux method, and multiblock method.

### 3.2. Simulation Setup and Verification

We changed the grid resolution for the analysis to verify the flow around flapping plates that imitate the body and pectoral fins of a fish. In this section, seven cycles of motion were conducted, and the physical quantities in the last four cycles were used to evaluate propulsive performance. In this section, the pectoral fins were set so that they did not rotate relative to the body and were always maintained at 20° relative to the body. Figure 12 shows the simulation area, which was 20 *L* × 10 *L* × 10 *L*, with a uniform flow of *u* = 62.5 mm/s in the inflow direction, a pressure gradient of 0, a reference pressure of 1/3 in Outflow 1 and a velocity gradient of 0. Outflow 2 had a pressure gradient of 0 and a velocity gradient of 0. Resolutions of 32, 64, and 128 cells/*L* were used. We simulate under the condition of *Re* = 500.

Thrust coefficient *C*_T_, power coefficient *C*_PWR_, and propulsion efficiency *η* were applied as evaluating parameters. They were obtained by these equations. Furthermore, each thrust coefficient for body and pectoral fins were obtained by nondimensionalizing each fluid force as shown in Equation (35).
(35)$$C_{T}=-\frac{F_{x,\mathrm{body}}+F_{x,\mathrm{fin}\_r}+F_{x,\mathrm{fin}\_l}}{\frac{1}{2}\rho U^{2}S_{p}},$$
(36)$$C_{\mathrm{PWR}}=-\frac{\sum_{l}^{\mathrm{surface}}u_{l,\mathrm{body}}\cdot f_{l,\mathrm{body}}+\sum_{l}^{\mathrm{surface}}u_{l,\mathrm{fin}\_r}\cdot f_{l,\mathrm{fin}\_r}+\sum_{l}^{\mathrm{surface}}u_{l,\mathrm{fin}\_l}\cdot f_{l,\mathrm{fin}\_l}}{\frac{1}{2}\rho U^{3}S_{p}},$$
(37)$$\eta =C_{T}\;/\;C_{\mathrm{PWR}},$$
where *F_x_* is the fluid force acting in the *x* direction in each model, ***u****_l_* and ***f****_l_* are the velocity and fluid force vectors in a small area in each model, and *S_p_* is the surface area of the computational model as in Section 3.1. As an evaluation index of the flow field, the *Q*-value was applied to extract vortices. The *Q*-value is obtained by the following equation:(38)$$\begin{array}{c} Q=\frac{1}{2}\left({\Omega_{\mathrm{ij}}}^{2}-{S_{\mathrm{ij}}}^{2}\right), \end{array}$$
where *S_ij_* is the deformation velocity tensor and *Ω_ij_* is the vorticity tensor. They can be obtained by
(39)$$\Omega_{\mathrm{ij}}=\frac{1}{2}\left(D_{\mathrm{ij}}-D_{\mathrm{ji}}\right),$$
(40)$$S_{\mathrm{ij}}=\frac{1}{2}\left(D_{\mathrm{ij}}+D_{\mathrm{ji}}\right),$$
where *D_ij_* is the velocity gradient tensor and is calculated as follows:(41)$$D_{\mathrm{ij}}=\left(\begin{array}{ccc} \frac{\partial u}{\partial x} & \frac{\partial u}{\partial y} & \frac{\partial u}{\partial z}\\ \frac{\partial v}{\partial x} & \frac{\partial v}{\partial y} & \frac{\partial v}{\partial z}\\ \frac{\partial w}{\partial x} & \frac{\partial w}{\partial y} & \frac{\partial w}{\partial z} \end{array}\right).$$

In this study, the *Q*-value and the vorticity *ω_x_* in the *x* direction were not dimensionalized as follows:(42)$$Q^{*}=\frac{Q}{{\left(U/L\right)}^{2}},$$
(43)$${\omega_{x}}^{*}=\frac{\omega_{x}}{U/L}.$$

Figure 13 shows the time history of the vorticity distribution *ω_x_** for the last half-cycle at a resolution of 128 cells/*L*. Figure 13 confirms that the formation of the vortex in Figure 13c–e and detachment of the vortex in Figure 13a,b,f can be confirmed due to the presence of the pectoral fins. The vortex of the fish body was separated at the upper and lower edges, and the pectoral fins caused the upper and lower vortex to split into two parts. In this analysis, because the pectoral fins did not rotate relative to the body, the vortices generated by the pectoral fins were small, and the effect of the pectoral fins on propulsive force was considered less dominant than that of the body.

Figure 14 shows the time histories of the thrust coefficient *C*_T_ at each grid resolution. As the number of grids per representative length increased, the high-frequency oscillations decreased and converged to a certain waveform. These high-frequency oscillations are nonphysical oscillations caused by objects crossing the lattice, and the effect can be reduced by increasing the lattice resolution. From Figure 13 and Figure 14, vortices were detached when thrust was high and vortex formation was encouraged when thrust was low. These results confirm the numerical reliability of the code. In the following results, the analysis was performed with a lattice resolution of 128 cells/*L*.

## 4. Results and Discussion

In this chapter, we evaluated the propulsive performance of the fish when the amplitude and frequency ratios of the pectoral fin and the phase between the pectoral fin and body were varied.

### 4.1. Effect of the Flapping Amplitude of Pectoral Fins

Under the condition that the pectoral fins and body move with the same periodicity, the analysis was carried out by changing the amplitude ratio of the pectoral fins from 0.0 to 0.2 in increments of 0.025. The movement history of the simulation model when the amplitude ratio was changed is shown in Figure 15, with the movement history of the model when the amplitude ratio is 0.1. Figure 16 shows the movement history when the amplitude ratio is 0.2. These results demonstrate that when the amplitude ratio was greater than 0.1, the robot stroked in the direction opposite to the body. In Figure 15, the absolute values of the angular velocity of the pectoral fins and that of the body were the same; therefore, the pectoral fins always maintained a constant angular translational motion in the Cartesian coordinate system. In this section, it should be noted that asymmetry motion was applied for pectoral fins. Here, the simulation area and conditions are presented as in the previous section. The amplitude ratio, frequency ratio, and phase difference between the body and pectoral fins in this analysis are shown in Table 1.

Figure 17 shows the vorticity distribution at the seventh cycle with an amplitude ratio *A*_fin_/*L*_fin_ of 0.20. Figure 17 shows that the vortex pectoral fins induced were much larger than those in Figure 13, and that the influence of the pectoral fins on the flow field increased as the amplitude increased.

Figure 18 shows the changes in the average thrust coefficient, the average power coefficient, and the propulsive efficiency during the last four cycles when the amplitude ratio varied. Figure 18a shows the average thrust coefficient of net value, body, right fin, and left fin. The net thrust coefficient decreased as the amplitude ratio increased. This result was due to the expansion of the projected area of the incoming fluid increasing as the amplitude ratio increased, and the increasingly dominant force received from the incoming fluid compared to the thrust produced by the pectoral fins. Higher thrust force of body and lower thrust force of fins were observed as amplitude ratio increased. Comparing Figure 17 with Figure 13, larger vortices were induced by pectoral fins, and they improved body thrust force as shown in Figure 18a. The reason why the thrust coefficient of fins decreased was the decreasing moving velocities of pectoral fins. In Figure 18b, the power coefficient decreased as the amplitude ratio increased. The decrease in pectoral fin velocity due to the increase in the amplitude ratio affected the amplitude, affecting the change in power. From the changes in the thrust and the power, the propulsive efficiency decreased as the amplitude ratio increased, as shown in Figure 18c. This suggests that there was a negative correlation between the amplitude ratio and propulsive performance under the condition of an amplitude ratio of 0.0 to 0.2.

### 4.2. Effect of the Flapping Frequency of Pectoral Fins

The analysis was performed by varying the frequency ratio of the pectoral fin to 0.25, 0.5, 1, 2, and 4 under the condition that the pectoral fins and body move with an amplitude ratio of 0.1. In this section, it should be noted that asymmetry motion was applied for pectoral fins for the frequency ratio of 1 and symmetry motion was applied for pectoral fins when the frequency ratio was not 1. Figure 19 shows the movement history of the model when the frequency ratio is two. The calculation area and conditions are the same as those of the previous section. The amplitude ratio, frequency ratio, and phase difference between the body and pectoral fins in this analysis are shown in Table 2. For this section, asymmetry motion was applied when frequency ratio was 1.

Figure 20 shows the vorticity distribution for the frequency ratio *f*_fin_/*f*_body_ of 4 in the seventh cycle. Figure 20 shows that the pectoral fins vortex was produced at a high frequency. The increased tip velocity of the pectoral fins caused the detachment of the vortex at the pectoral fin edge, similar to that in the body, confirming that two vortices were generated. In contrast, the vortex became smaller when it reached near the tail of the body compared to the state vortex with an increased amplitude ratio. The amplitude ratio had a stronger effect on the hydrodynamic interaction between the pectoral fins and the movement of the fish body.

Figure 21 shows the changes in the average thrust coefficient, power coefficient, and propulsive efficiency during the last four cycles for different frequency ratios. Figure 21a suggests that the net thrust coefficient and the thrust coefficient of pectoral fins increased as the frequency ratio became higher than 1. The body thrust coefficient decreased as the frequency ratio became higher than 1. As we can see from Figure 20, vortexes were produced at a high frequency and improved the thrust coefficient of pectoral fins. When the frequency ratio became lower than 1, the net thrust coefficient and the thrust coefficient of pectoral fins increased as the frequency ratio decreased. The body thrust coefficient decreased as the frequency ratio decreased. These results represent that as the frequency ratio decreased, the relative velocity of pectoral fins became lower, and the velocity of pectoral fins became higher. Therefore, the thrust coefficient of pectoral fins became higher. Figure 21b also shows a slight increase in the power coefficient with increasing frequency ratio. As shown in Figure 21c, propulsive efficiency increased as the frequency ratio became larger than 1, and propulsive efficiency increased as the frequency ratio became lower than 1. This indicates that the frequency ratio is an effective parameter for significantly increasing thrust under the present conditions.

### 4.3. Effect of Phase Difference between Pectoral Fins and Fish Body

The phase difference between the pectoral fins and the body of the fish was varied from 0° to 360° in increments of 30°. Figure 22 shows the movement history when the frequency ratio is 2 and the phase difference is 180°. In this section, it should be noted that asymmetry motion was applied for pectoral fins for the frequency ratio of 1 and symmetry motion was applied for pectoral fins when the frequency ratio was not 1. The computational domain and conditions are the same as those of the previous section. The amplitude ratio, frequency ratio, and phase difference between the body and pectoral fin in this analysis are shown in Table 3.

Figure 23 shows the vortex time history for a phase difference of 180° and a frequency ratio of 2. From this figure, the tip speed of the pectoral fins was faster than the state for the phase of 0° when the frequency ratio was 2 and the phase was 180°. Thus, the vortex generated by the pectoral fins was therefore larger, and its effect on the flow field was considered to be greater. As in the case of a frequency ratio of 4, when the pectoral fins are moving rapidly, two vortices are generated by the separation of the upper and lower edges of the pectoral fin vortex.

Figure 24 shows the changes in the average thrust coefficient, the average power coefficient, and the propulsive efficiency for the last four cycles as the phase difference varied. In Figure 24a, when the frequency ratio was 1, the highest thrust was generated when the phase difference was 120°, and the thrust coefficient changed with the phase difference in an upward convex shape. This may be because the pectoral fin velocity was small when the phase difference was 0° because the angle did not change in the Cartesian coordinate system, and the pectoral fin velocity increased significantly when the phase difference was set. As the frequency ratio increased, the effect of the phase difference decreased and the change in the thrust coefficient decreased. This is likely because the pectoral fin tip velocity increased as the frequency ratio increased, and the change in the tip velocity due to the phase difference became relatively small. Figure 24b represents the change of each thrust coefficient for the frequency ratio of 1. This figure shows that phase difference effected to thrust forces of the body and pectoral fins, and they indicated a higher coefficient for a phase difference of around 90° and a lower coefficient for a phase difference of around 270°. Similarly, in Figure 24c, the effect of phase difference decreased as the frequency ratio increased. When the frequency ratio was 1, the power was smallest when the phase difference was 90°. The power coefficient change was similar to the thrust coefficient change. It was found that fluid force was more dominant than the change of tip velocity of pectoral fins to change the power coefficient. As shown in Figure 24d, except for the phase difference of 0° and larger than 240°, the propulsive efficiency at a frequency ratio of 1 was superior to that at a frequency ratio of 2 or higher. The highest propulsion efficiency increased by a factor of about 3.7 compared to the phase difference condition of 0. Figure 24c shows that the power was also lower when the frequency ratio was higher. This indicates that changing the phase difference may greatly improve the propulsive performance when the frequency ratio is 1. The effect of phase difference on propulsive performance was small for frequency ratios greater than 2.

## 5. Conclusions

In order to investigate the effect of pectoral fin motion on swimming performance in a rigid plate model of a fish with pectoral fins, we compared the propulsive efficiency of the model when the amplitude, frequency, and phase difference of the pectoral fins were varied. The results are as follows:When the amplitude ratio was increased, the coefficients of thrust and power tended to decrease. Propulsive efficiency also tended to decrease as the amplitude ratio increased. This may be due to the large projected area of the pectoral fin, and the drag force due to the inflow becoming larger. The increase in the amplitude of the pectoral fin vortex was larger, indicating that the influence of the pectoral fin motion on the flow field was increased.The thrust coefficient, power coefficient, and propulsive efficiency increased when the pectoral fin frequency was increased. This may be due to the change in the projected area being smaller than when the amplitude ratio was changed, whereas the increase in the pectoral fin tip velocity with increasing frequency increased the thrust produced by the pectoral fin. This finding was confirmed from the flow field that the increase in frequency caused the pectoral fin vortex to form a vortex train, which facilitated propulsion by the pectoral fins.When the frequency ratio was one, the highest propulsive efficiency was observed at a phase difference of 120°, which showed the best propulsive performance compared with other frequency ratios. When the frequency ratio ≥ 2, the change due to the phase difference was small, and it was confirmed that the effect of the phase difference decreased as the frequency ratio became increased.

## Figures and Tables

**Figure 1 biomimetics-09-00156-f001:**
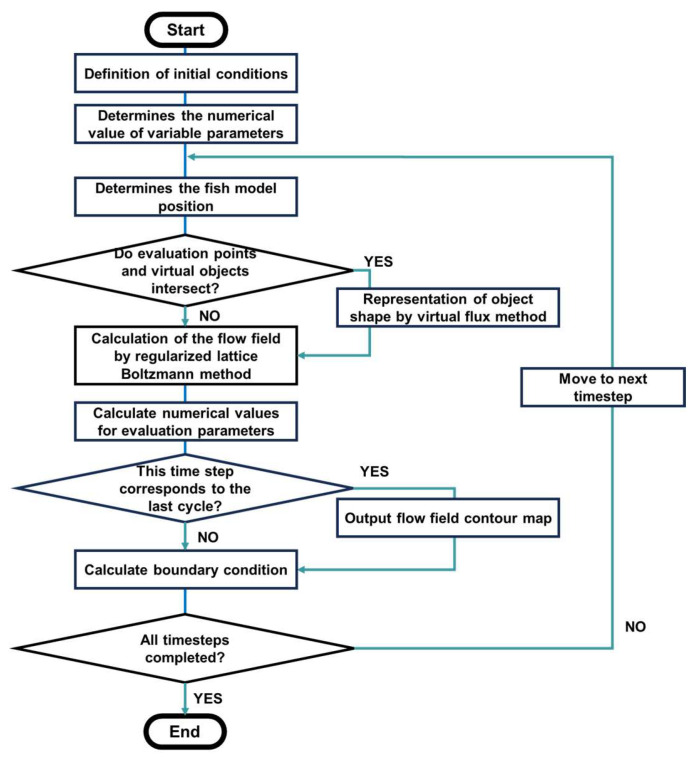
Flow chart for the simulation of fish swimming.

**Figure 2 biomimetics-09-00156-f002:**
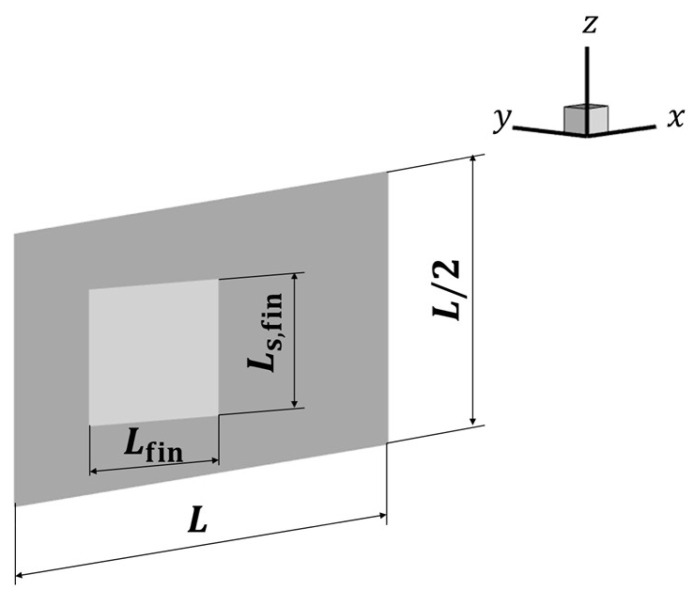
Computational model of the body and pectoral fins.

**Figure 3 biomimetics-09-00156-f003:**
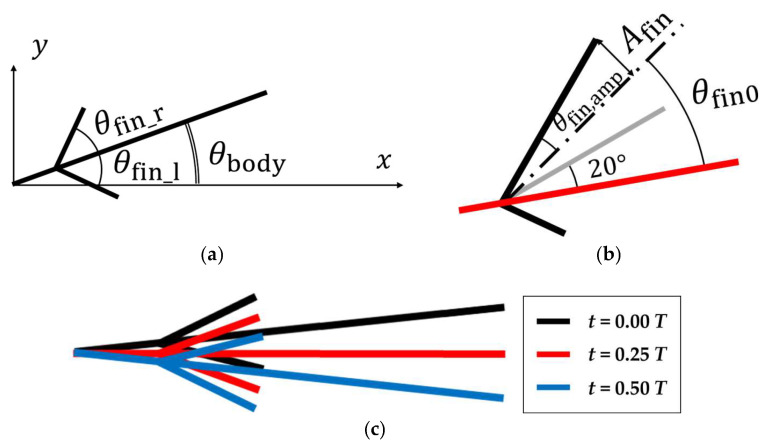
Schematic view. (**a**) Angles of body and pectoral fins; (**b**) the relative angle of the right fin; (**c**) time history of the motion of calculation model.

**Figure 4 biomimetics-09-00156-f004:**
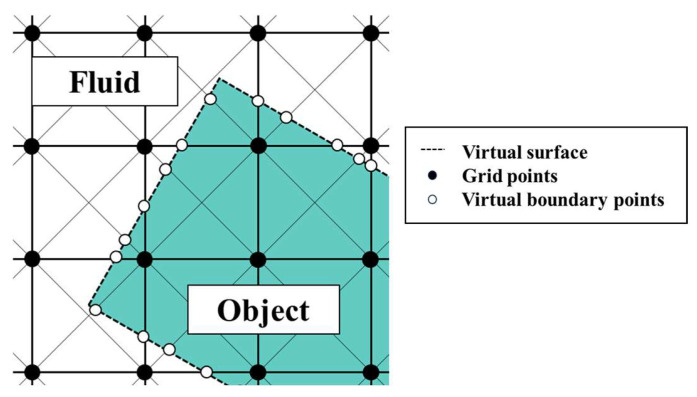
Schematic view of the virtual boundary points in the virtual flux method.

**Figure 5 biomimetics-09-00156-f005:**
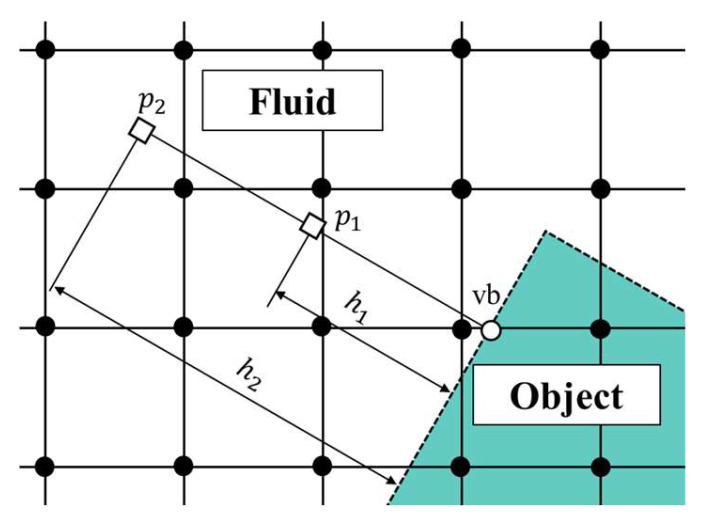
Extrapolation on the virtual boundary surface.

**Figure 6 biomimetics-09-00156-f006:**
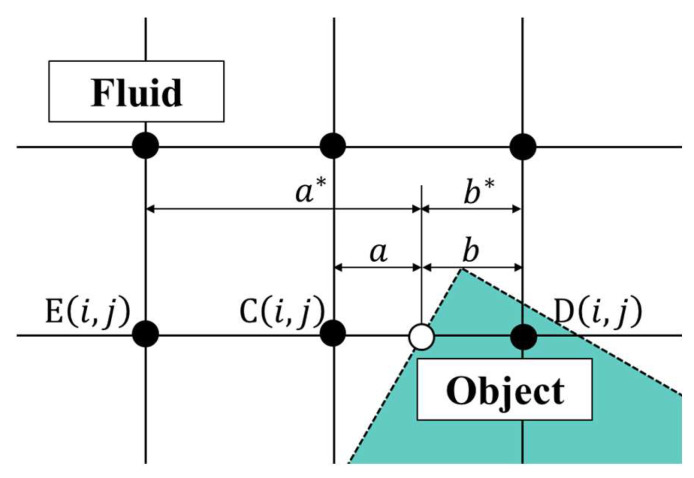
Schematic view of virtual boundary points where *a* and *b* are the interior division of line segment CD and *a** and *b** are the interior division of line segment ED.

**Figure 7 biomimetics-09-00156-f007:**
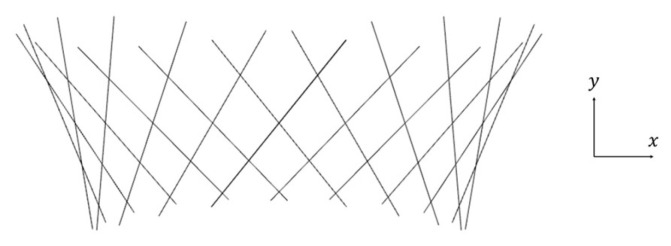
A schematic of the movement of a flapping plate.

**Figure 8 biomimetics-09-00156-f008:**
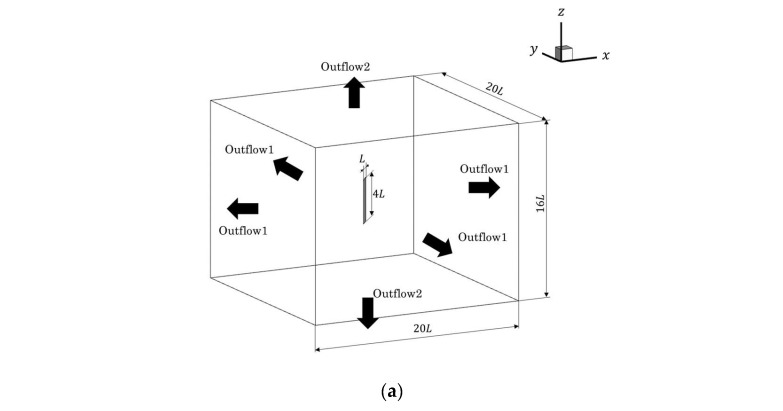
Schematic views of the simulation area for the flow around a flapping plate: (**a**) Bird’s eye view; (**b**) *x*–*y* plane; (**c**) *y*–*z* plane.

**Figure 9 biomimetics-09-00156-f009:**
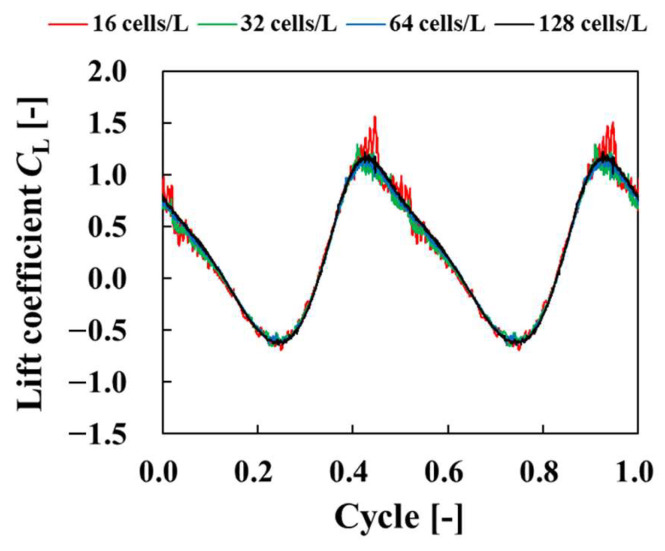
Time histories of lift coefficients for flow around a flapping plate.

**Figure 10 biomimetics-09-00156-f010:**
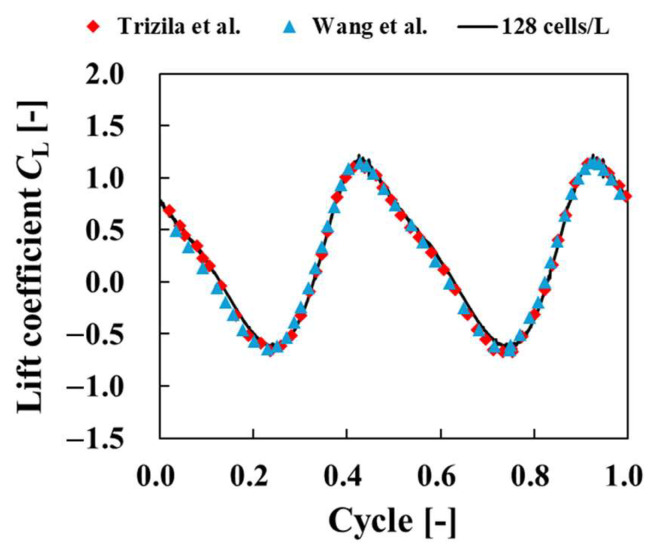
Time histories of lift coefficients for flow around a flapping plate compared to computational results of Trizilla [32] and Wang et al. [33].

**Figure 11 biomimetics-09-00156-f011:**
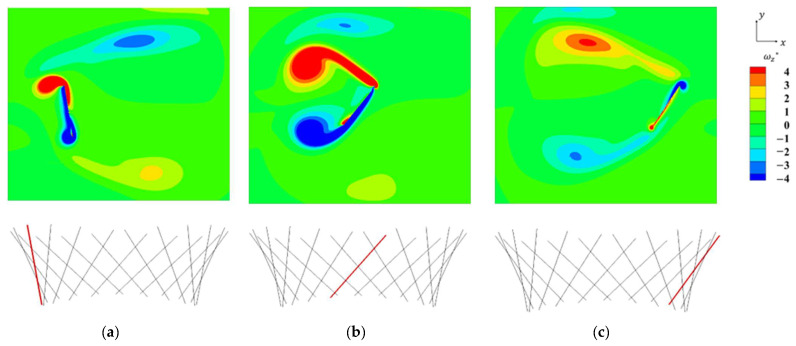
The *z*-vorticity contours around the midspan of a flapping plate at selected time instants during the forward stroke: (**a**) *t* = 6.8*T*; (**b**) *t* = 6.0*T*; (**c**) *t* = 6.2*T*.

**Figure 12 biomimetics-09-00156-f012:**
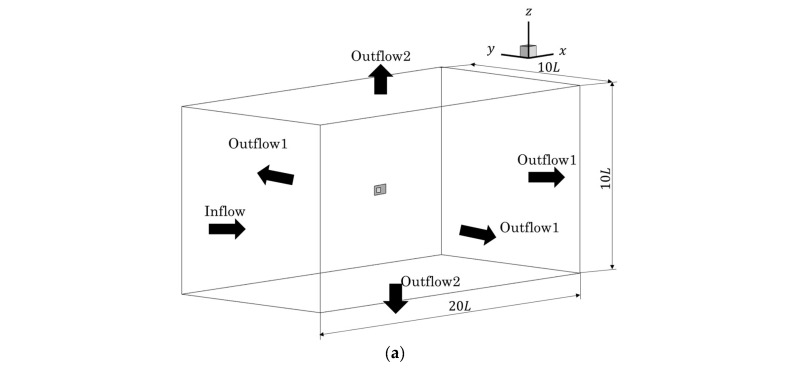
Schematic views of the simulation area for the flow around a flapping plate: (**a**) Bird’s eye view; (**b**) *x*–*y* plane; (**c**) *x*–*z* plane.

**Figure 13 biomimetics-09-00156-f013:**
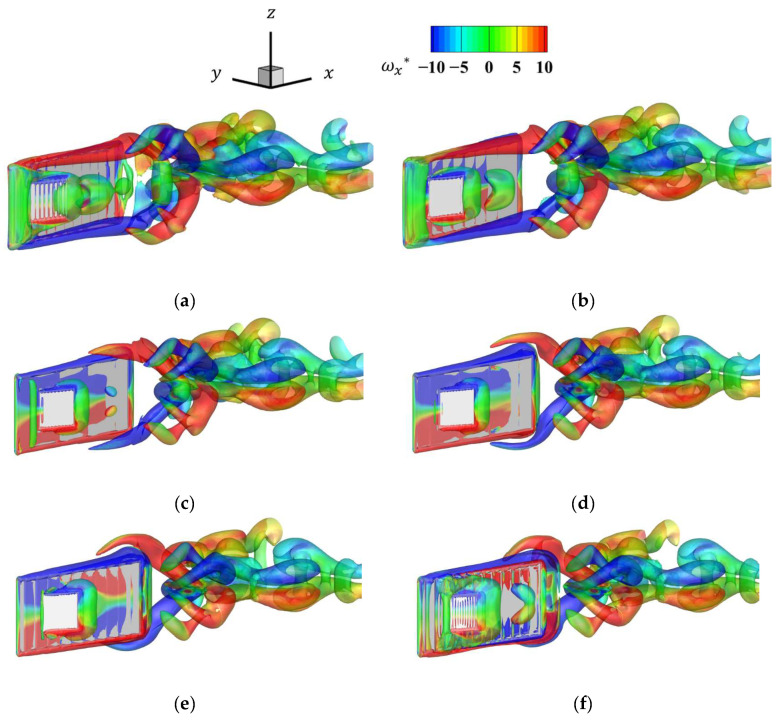
Time history of the *x* directional vorticity distribution for the last half-cycle: (**a**) *t* = 0.0*T*; (**b**) *t* = 0.1*T*; (**c**) *t* = 0.2*T*; (**d**) *t* = 0.3*T*; (**e**) *t* = 0.4*T*; (**f**) *t* = 0.5*T*.

**Figure 14 biomimetics-09-00156-f014:**
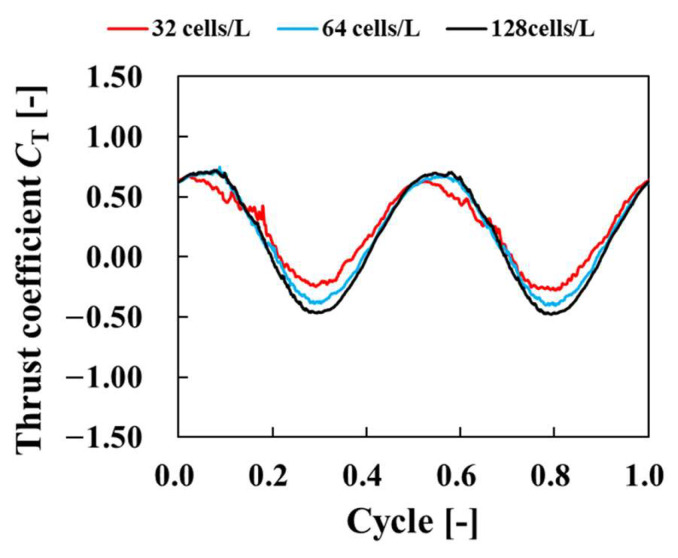
Time history of thrust coefficient for each number of grids.

**Figure 15 biomimetics-09-00156-f015:**
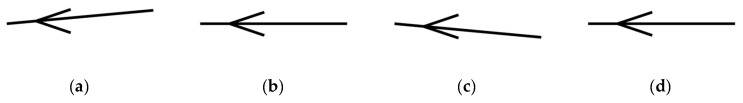
The movement history in one cycle for amplitude ratio *A*_fin_ ⁄ *L*_fin_ of 0.10: (**a**) *t* = 0.00*T*; (**b**) *t* = 0.25*T*; (**c**) *t* = 0.50*T*; (**d**) *t* = 0.75*T*.

**Figure 16 biomimetics-09-00156-f016:**
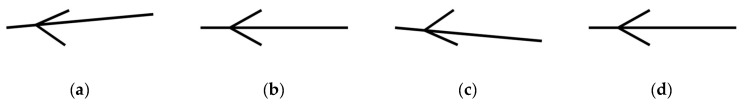
The movement history in one cycle for amplitude ratio *A*_fin_ ⁄ *L*_fin_ of 0.20: (**a**) *t* = 0.00*T*; (**b**) *t* = 0.25*T*; (**c**) *t* = 0.50*T*; (**d**) *t* = 0.75*T*.

**Figure 17 biomimetics-09-00156-f017:**
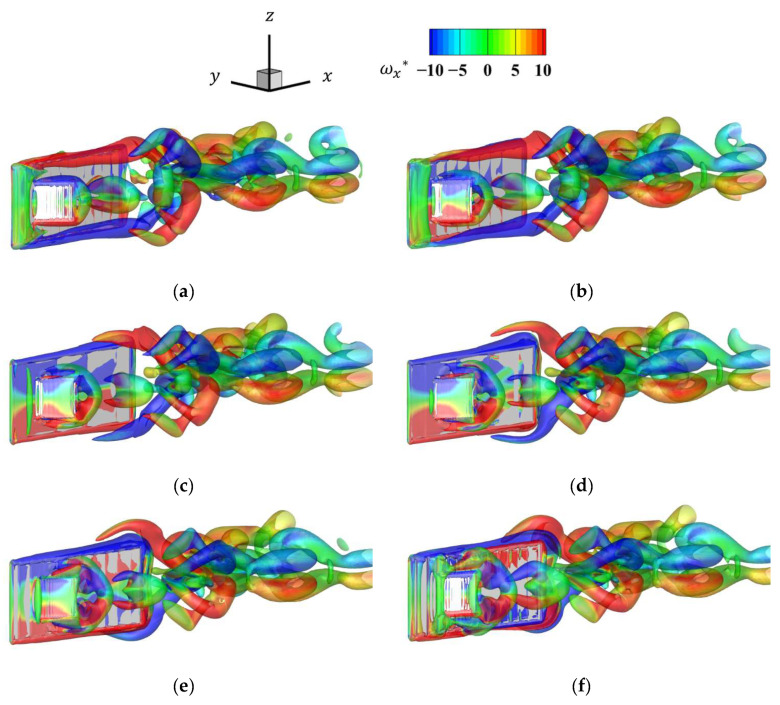
Time history of the *x* directional vorticity distribution for the last half-cycle for an amplitude ratio of 0.2: (**a**) *t* = 0.0*T*; (**b**) *t* = 0.1*T*; (**c**) *t* = 0.2*T*; (**d**) *t* = 0.3*T*; (**e**) *t* = 0.4*T*; (**f**) *t* = 0.5*T*.

**Figure 18 biomimetics-09-00156-f018:**
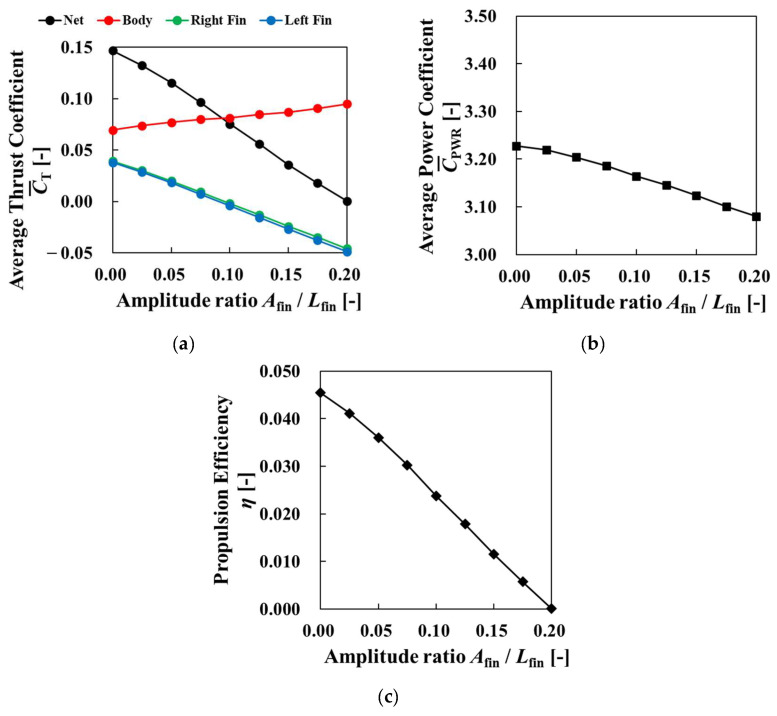
Average thrust coefficient, power coefficient, and propulsion efficiency for four cycles versus the amplitude ratio of the pectoral fins: (**a**) average thrust coefficient; (**b**) average power coefficient; (**c**) propulsion efficiency.

**Figure 19 biomimetics-09-00156-f019:**
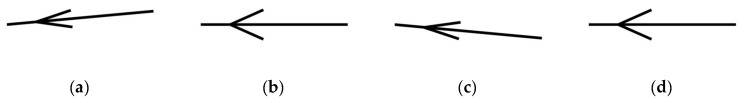
Movement history in one cycle for the frequency ratio *f*_fin_ ⁄ *f*_body_ of 2: (**a**) *t* = 0.00*T*; (**b**) *t* = 0.25*T*; (**c**) *t* = 0.50*T*; (**d**) *t* = 0.75*T*.

**Figure 20 biomimetics-09-00156-f020:**
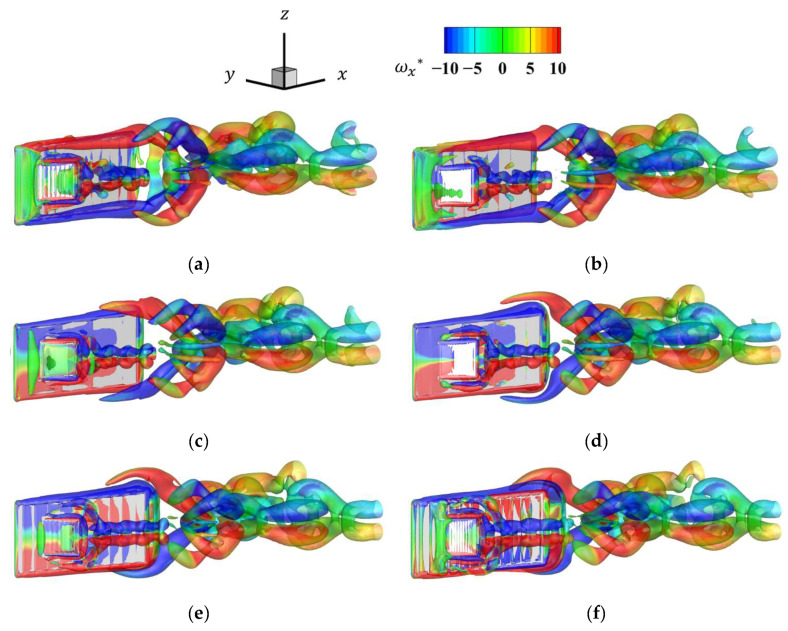
Time history of the *x*-directional vorticity distribution for the last half-cycle for a frequency ratio of 4: (**a**) *t* = 0.0*T*; (**b**) *t* = 0.1*T*; (**c**) *t* = 0.2*T*; (**d**) *t* = 0.3*T*; (**e**) *t* = 0.4*T*; (**f**) *t* = 0.5*T*.

**Figure 21 biomimetics-09-00156-f021:**
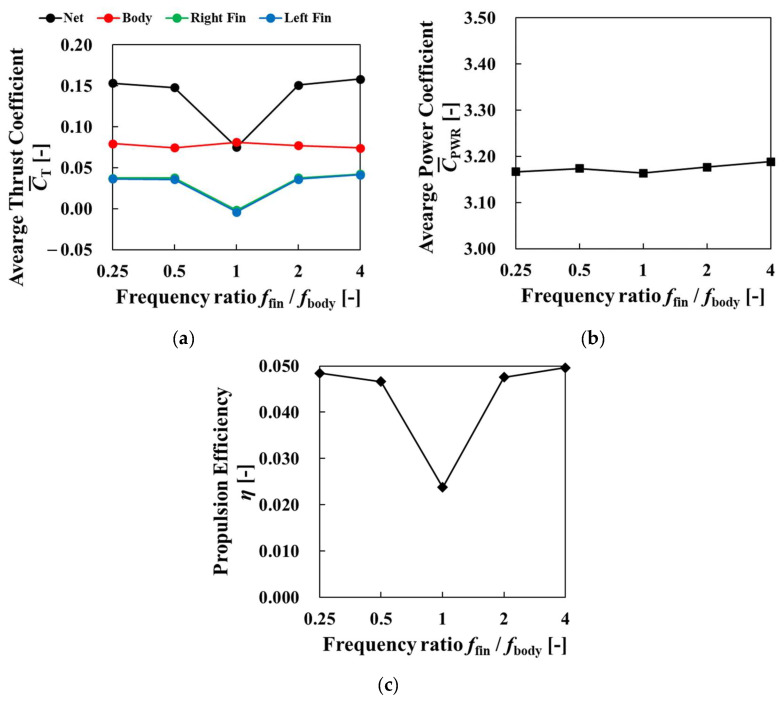
Average thrust coefficient, power coefficient, and propulsion efficiency for four cycles versus frequency ratio: (**a**) average thrust coefficient; (**b**) average power coefficient; (**c**) propulsion efficiency.

**Figure 22 biomimetics-09-00156-f022:**
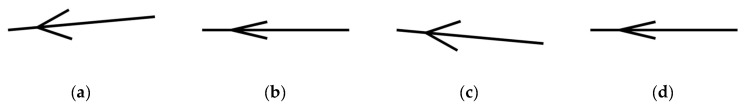
Movement history in one cycle for the frequency ratio *f*_fin_ ⁄ *f*_body_ of 2 and phase *φ* of 180°: (**a**) *t* = 0.00*T*; (**b**) *t* = 0.25*T*; (**c**) *t* = 0.50*T*; (**d**) *t* = 0.75*T*.

**Figure 23 biomimetics-09-00156-f023:**
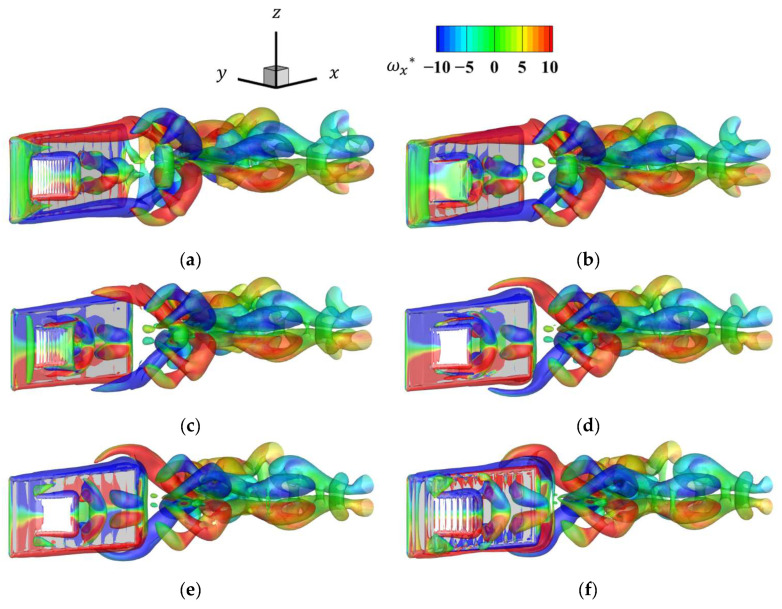
Time history of the *x*-directional vorticity distribution for the last half-cycle for a frequency ratio of 2 and a phase of 180°: (**a**) *t* = 0.0*T*; (**b**) *t* = 0.1*T*; (**c**) *t* = 0.2*T*; (**d**) *t* = 0.3*T*; (**e**) *t* = 0.4*T*; (**f**) *t* = 0.5*T*.

**Figure 24 biomimetics-09-00156-f024:**
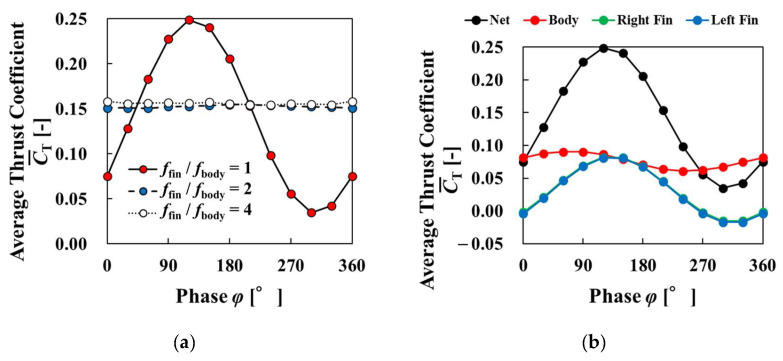
Average thrust coefficient, power coefficient, and propulsion efficiency for four cycles versus phase difference for different frequency ratios: (**a**) average thrust coefficient; (**b**) average thrust coefficient for each organ with frequency ratio of 1; (**c**) average power coefficient; (**d**) propulsion efficiency.

**Table 1 biomimetics-09-00156-t001:** Parameter definition for changing the amplitude ratio.

Amplitude ratio of pectoral fins	*A*_fin_/*L*_fin_ [-]	0.000, 0.025, 0.050, 0.075, 0.100, 0.125, 0.150, 0.175, 0.200
Frequency ratio	*f*_fin_/*f*_body_ [-]	1
The phase between body and fins	*φ* [°]	0

**Table 2 biomimetics-09-00156-t002:** Definition of parameters to change the frequency ratio.

Amplitude ratio of pectoral fins	*A*_fin_/*L*_fin_ [-]	0.100
Frequency ratio	*f*_fin_/*f*_body_ [-]	0.25, 0.5, 1, 2, 4
Phase between body and fins	*φ* [°]	0

**Table 3 biomimetics-09-00156-t003:** Parameter definition for the changing phase.

Amplitude ratio of pectoral fins	*A*_fin_/*L*_fin_ [-]	0.100
Frequency ratio	*f*_fin_/*f*_body_ [-]	1, 2, 4
Phase between body and fins	*φ* [°]	0, 30, 60, 90, 120, 150, 180, 210, 240, 270, 300, 330, 360

## Data Availability

Dataset available on request from the authors.

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
