# Peer review of "Three-Dimensional Numerical Study of Hydrodynamic Interactions between Pectoral Fins and the Body of Aquatic Organisms"

_biomimetics, 2024, doi:10.3390/biomimetics9030156_

Round 1
Reviewer 1 Report
Comments and Suggestions for Authors
This work numerically studied the hydrodynamic interaction between pectoral fins and fish bodies using three-dimensional Lattice Boltzmann Method. Many investigation of the effects of amplitude ratio、frequency ratio and phase of the body were carried out in this study that it is an interesting and sound investigation. I do have some questions and would recommend the manuscript to be accepted with major revisions.
My comments are as follows:
1. Why does the motion equation of the body adopt the sine and cosine equation of oscillation? Can the author describe the envelope diagram of the movement of the section or the shape of the trajectory?
2. Is the equation of motion starting from the head? The fitting coordinates are suggested in the schematic diagram.
3. The equation include motion in the x direction, so is it a self-propelled motion pattern?
4. Why the amplitude ratio Abody/L was fixed at 0.1?
5. What is the definition of dfin in the formula 4~7?
6. Why the definition of U in formula 9 is not the inflow speed. And if it is the average velocity of the body of the tail. A more detailed definition is needed.
7. The Q-value is defined in the paper. Why is it not used in the flow field description?
8. The asymmetry between the left and right fins will affect the maneuvering performance, does it need to consider the impact of Torque Mz?
9. It is suggested to put 3.1 and 3.2 in a separate part and put Result in the fourth part.
10. What is the phase difference between the left and right fin in original section 3.3? This needs to be clearly stated.
11. Is it possible to isolate the influence of each part of the body, left fin and right fin? It may help to gauge the size of the interference between the left and right fins and the body.
12. The sample space of frequency ratio is too small. More cases are needed.
13. The law and the mechanism behind the flow field need to be more in-depth analysis.
Comments on the Quality of English LanguageMinor editing of English language required.
Reviewer 2 Report
Comments and Suggestions for Authors
The article entitled " Three-dimensional Numerical Study of Hydrodynamic Interactions between Pectoral Fins and the Body of Aquatic Organisms” is carefully reviewed and it is advised to address the following comments before accepting the article.
The abstract should be modified with some important outcomes in quantitative terms.
A short and clear objective and uniqueness of the research must be added at the end of the introduction.
It advised adding a minimum of 4 to 5 recent literature i.e 2023/2024 to mitigate the research topic is up to date.
The Section 2 Numerical Method should be replaced with materials and methods.
Any materials adopted should be mentioned under section 2.1 Materials.
All the methods follow 2.2, 2.3, etc.
A flow chart must be added after 2.1 for easy understanding of the method adopted.
If any equations is taken from the literature, it should be duly cited.
The Figure 11 explanation needs to elaborate, by explaining the important parameters obtained from the contour map.
The Figure 13 and 14 sub figures i.e Figure 13a, 13b…, 14a, 14b all need to explain.
Figure 23 shows the vortex time history for a phase difference of 180° and a frequency ratio of 2. Figure 23 shows that the tip speed of the pectoral fins was faster than the state for the phase of 0° when the frequency ratio was 2 and the phase was 180°. It is observed that there is a repetition of “Figure 23 shows”, so modify the sentence accordingly without any repetition of similar words.
There are a lot of grammatical and sentence connection errors observed, check and modify accordingly
Comments on the Quality of English LanguageThere are a lot of grammatical and sentence connection errors observed, check and modify accordingly
Reviewer 3 Report
Comments and Suggestions for Authors
The article is devoted to numerical study of fish swimming. The problem is interesting and practically important because fish locomotion system woks with excellent propulsive efficiency.
It is quite obvious, the fishes swim by moving their body, fins, and other organs simultaneously, which developed during evolution. In particular, the pectoral fin plays a crucial role in swimming. It works for forward-backward movement and change of direction.
The authors study the model where a motion was performed in the x-y plane, and no motion in the z direction was assumed. The model outline is shown in Fig.1 of the manuscript. It has a specially simplified for the simulation convenience shape of the pectoral fins. To solve dynamic equations they use so-called virtual flux method which allows represent objects of arbitrary shape on a Cartesian grid.
A schematic view of the virtual boundary points in the virtual flux method is shown in Fig. 4 of the manuscript. Briefly, the virtual boundary points were placed at the intersection of the object surface and the discrete velocity direction, as shown by the white dots in the mentioned Fig. 4. Further the perform an extrapolation on the virtual boundary surface and apply it for the practical calculation.
The results are basically summarized in Figs. 10-14. One has to note beautiful 3d plots with time consuming histories calculated for the vorticity distribution. The authors extract from the simulations and plot the following important information: average thrust coefficient, power coefficient, and propulsion efficiency for four-cycles versus the amplitude ratio of the pectoral fins accumulated in the subplots (a)-(c) of Fig.18. Besides they find average thrust coefficient, power coefficient, and propulsion efficiency for four-cycles versus frequency ratio shown in Fig 21.
All the announced by authors goals related to the propulsive efficiency are reached, and corresponding results obtained in the frames of their model at varied: amplitude, frequency and phase difference of the pectoral fins were briefly, but completely summarized in the “Conclusions”.
The idea of the study is absolutely clear. The simulations look as done professionally and properly described. The literature in the field is sufficiently cited. So, I can recommend the publication of the paper in its present form.
Round 2
Reviewer 1 Report
Comments and Suggestions for Authors This work numerically studied the hydrodynamic interaction between pectoral fins and fish bodies using three-dimensional Lattice Boltzmann Method.Many investigation of the effects of amplitude ratio、frequency ratio and phase of the body were carried out in this study that it is an interesting and sound investigation.
The author responded well to my comments and made reasonable revisions. So I agree to recommend this article for publication in the journal.
Reviewer 2 Report
Comments and Suggestions for Authors
As the authors incorporated the comments, hence it may be accepted now for publication.
Comments on the Quality of English LanguageAs the authors incorporated the comments, hence it may be accepted now for publication.